# Differentially Private Block Coordinate Descent for Linear Regression on Vertically Partitioned Data

**Jins de Jong** \*,†, **Bart Kamphorst** † **and Shannon Kroes** †

TNO, Unit ICT, Eemsgolaan 3, 9727DW Groningen, The Netherlands
* Correspondence: jins.dejong@tno.nl
† Alphabetically ordered list of authors.

**Abstract:** We present a differentially private extension of the block coordinate descent algorithm by means of objective perturbation. The algorithm iteratively performs linear regression in a federated setting on vertically partitioned data. In addition to a privacy guarantee, we derive a utility guarantee; a tolerance parameter indicates how much the differentially private regression may deviate from the analysis without differential privacy. The algorithm's performance is compared with that of the standard block coordinate descent algorithm on both artificial test data and real-world data. We find that the algorithm is fast and able to generate practical predictions with single-digit privacy budgets, albeit with some accuracy loss.

**Keywords:** differential privacy; federated learning; vertically partitioned data





## 1. Introduction

There are many circumstances where organizations need to use each other's data to perform tasks, such as data analysis or prediction [1–3]. For example, different parties may own different data sets that can be combined for improved predictive performance or inference. When these data contain personal information, the required data exchange can be problematic. In these situations, federated learning can be used to facilitate such collaborations. It is a privacy-preserving technique that keeps the data local during the analysis and ensures that no other party gains access to it. Within the research field dedicated to federated learning, there is increasing attention towards solutions for vertically partitioned data. One speaks of vertically partitioned data when different parties owning different attributes on the same subjects.

An analysis that researchers often seek to perform on vertically partitioned data is regression analysis. Recently, an approach has been presented for this scenario: Block coordinate descent (BCD) [4]. BCD is a promising and fast way to perform federated learning for generalized linear models. One of its strengths is that it avoids computationally expensive cryptographic operations to secure the computations. Although no raw data is shared during BCD, the information that is exchanged can still leak information, as no privacy guarantees are in place. There are several examples of this in the context of federated learning [5,6]. To limit the possibility of information leakage, we supplement BCD with differential privacy (DP).

Differential privacy was introduced by [7] and is widely referred to as the state-of-the art approach to privacy preservation. Essentially, it involves adding uncertainty to the data analysis such that similar data sets will likely lead to similar results. Since its introduction, differential privacy has observed some application, but not yet widespread adoption. One of the reasons for this is that the DP parameters quantifying the privacy guarantees often cannot be made as small as hoped while preserving utility. The result is a noisy learning algorithm with reduced performance that theoretically could reveal information about data in the data set with considerable certainty.

The motivation for this project is twofold. The first is to extend BCD with privacy guarantees to make it applicable in a wider range of use cases. The second motivation is to improve the practicality of DP in realistic use cases. To do so, we make some optimistic choices in our set-up. This means that less noise has to be added and a better performance is obtained. This clearly reduces the amount of protection DP offers. However, we believe that in this way we provide more meaningful privacy guarantees that correspond better to the data analyst's practice.

*Related Work*

Multiple approaches have been presented to add differential privacy to a linear regression problem in the centralized setting (i.e., with one party) [8–11]. For a federated setting, there has been more focus on horizontally partitioned data [12,13]. For vertically partitioned data, only a few solutions are known [14,15]. A problem similar to ours is treated in [15], where it is approached using techniques from multi-party computation. Although the algorithm performs well, it does not provide a utility guarantee or expectation.

Utility expectations of differentially private learning have been presented for other applications. Examples can be found in differentially private empirical risk minimization [8,16–18]. Since such utility bounds are typically asymptotic, large numbers of iterations are required for such bounds to become reliable. For learning techniques with a small number of iterations, such as ours, these are not practical.

The approaches above apply differential privacy over the entire universe of data sets. The concept of locally sensitive differential privacy has been studied before [19–21], albeit under various names.

*Our Contributions*

We introduce DP-BCD, a slightly reformulated version of the block coordinate descent algorithm [4] that has been made differentially private (DP) using objective perturbation [8,22–24]. It iteratively performs linear regression on vertically partitioned data. To make this implementation as practical as possible, we use local sensitivity parameters in a particularly small universe of possible data sets, instead of using global upper bounds on some large set of unseen data sets. Furthermore, we introduce a new parameter $\gamma$ that gives the maximally alowable performance decay per iteration. Before the analysis, the parties agree on a loss scaling, fixing the amount of performance they are willing to sacrifice for more privacy. As a consequence of this, theoretical performance guarantees can be derived. To evaluate its performance, we compare our algorithm with the standard BCD algorithm without differential privacy on synthetic test data, the forest fires data set [25] and the garments industry data [26].

*Outline*

In Section 2, we formulate the federated setting and introduce the fundamental results from regression analysis and differential privacy that we need for our construction. The construction of the DP-BCD algorithm with the main result, Theorem 1, can be found in Section 3.1. In Section 3.2, we compare the performance of DP-BCD with standard BCD and linear regression in the centralized setting. In Section 4, we discuss its performance and some improvements of the algorithm, and we conclude with Section 5.

## 2. Materials and Methods

This section elaborates on the federated context that motivates and scopes our research. Thereafter, we present the BCD algorithm for training a simple linear regression model and highlight the potential privacy issues. Finally, as a stepping stone for improving the BCD algorithm, we formally introduce differential privacy.

### 2.1. Federated Context

Federated learning with $k$ parties involves local data sets $\{\boldsymbol{X}^{(j)} | 1 \leq j \leq k\}$ that jointly form a federated data set $(\boldsymbol{X}^{(1)}, \dots, \boldsymbol{X}^{(k)})$. These data sets are used to jointly train a model,

which in this case consists of the joint weights $(\boldsymbol{\beta}_a, \boldsymbol{\beta}_b)$. The essence of federated learning is that the local data set of any party is only accessed by the party itself, ensuring that no other party processes it. So federated learning can be chosen to provide more data confidentiality. However, information about the local data set may still be deduced from the outcomes of the local computation.

Our setup assumes that the data sets and the list of participants is fixed for the entire runtime of the algorithm. Nonetheless, modifications that allow the addition of new subjects or objects to the data sets are conceivable. It may also be possible to add a participant during the protocol. This participant will simply have missed the first few iterations and have not contributed anything there. Participants cannot stop during the protocol without publishing their result so far. Such extensions are out of scope for this work. It should be clear that the results here all assume a fixed list of participants and data sets.

In the rest of the article, we will work in the two-party setup. The algorithms and results can be generalized to the *k*-party setting in a straightforward manner. The utility results do depend on the number of parties. We assume that two entities, named Alice and Bob, intend to perform an analysis on their joint tabular data. These entities could be researchers, analysts or some organizations. The data is vertically partitioned, meaning that Alice and Bob have complementary data on the same subjects. More specifically, the data $\boldsymbol{X}_a$ of Alice and the data $\boldsymbol{X}_b$ of Bob, with respective dimensions $N \times m_a$ and $N \times m_b$, both contain $N$ observations that are ordered in the same way. Alice knows the first $m_a$ parameters of each observation and Bob knows the other $m_b = m - m_a$ observations. In this setting, the methods for linear regression in the centralized setting [8] provide good differentially private linear regression algorithms.

### 2.2. Linear Regression

We consider simple linear regression, which is the problem of finding $\boldsymbol{\beta}^*$, such that

$$\mathcal{L}(\boldsymbol{\beta}^*) = \min_{\boldsymbol{\beta}} \mathcal{L}(\boldsymbol{\beta}), \tag{1}$$

where the loss $\mathcal{L}$ on the the data set $\boldsymbol{X}$ with labels $\boldsymbol{y}$ is given by

$$\mathcal{L}(\boldsymbol{\beta}) := \|\boldsymbol{X}\boldsymbol{\beta} - \boldsymbol{y}\|_2^2. \tag{2}$$

The optimal solution to this problem is found by deriving with respect to $\boldsymbol{\beta}$ and determining its root, yielding

$$\boldsymbol{\beta}^* = (\boldsymbol{X}^T \boldsymbol{X})^{-1} \boldsymbol{X}^T \boldsymbol{y}. \tag{3}$$

### 2.3. Block Coordinate Descent

The starting point is the block coordinate descent (BCD) algorithm introduced by [4]. It can be used to train a generalized linear model in a federated setting. In the standard approach, all parties know the label $\boldsymbol{y}$. The first party tries to create a linear model to predict as much from $\boldsymbol{y}$ as possible from its own data. It hands its prediction to the next player, who tries to improve the prediction as much as possible. This continues until the stopping criterion is met.

A simple modification of the original algorithm communicates the missing parts rather than their own predictions. This has the advantages that only a single party needs to know the true label. This is a common situation in many joint learning problems. For this reason the *single label owner* variant Algorithm 1 of BCD will be used here.

---

**Algorithm 1** Incremental 2-party block coordinate descent algorithm. The subscript $a$ is for Alice and $b$ for Bob

---

1:  Alice and Bob initiate $\boldsymbol{\beta}_a^{(0)} \leftarrow \mathbf{0}$ and $\boldsymbol{\beta}_b^{(0)} \leftarrow \mathbf{0}$, respectively
2:  Alice initiates $\boldsymbol{v}_b \leftarrow \boldsymbol{y}$
3:  $i \leftarrow 0$
4:  **while** stopping criterion is not met **do**
5:      **player** Alice **do**
6:          $\tilde{\boldsymbol{\beta}}_a \leftarrow (\boldsymbol{X}_a^T \boldsymbol{X}_a)^{-1} \boldsymbol{X}_a^T \boldsymbol{v}_b$
7:          $\boldsymbol{\beta}_a \leftarrow \boldsymbol{\beta}_a + \tilde{\boldsymbol{\beta}}_a$
8:          $\boldsymbol{v}_a \leftarrow \boldsymbol{v}_b - \boldsymbol{X}_a \tilde{\boldsymbol{\beta}}_a$
9:          send $\boldsymbol{v}_a$ to Bob
10:    **end player**
11:    **player** Bob **do**
12:        $\tilde{\boldsymbol{\beta}}_b \leftarrow (\boldsymbol{X}_b^T \boldsymbol{X}_b)^{-1} \boldsymbol{X}_b^T \boldsymbol{v}_a$
13:        $\boldsymbol{\beta}_b \leftarrow \boldsymbol{\beta}_b + \tilde{\boldsymbol{\beta}}_b$
14:        $\boldsymbol{v}_b \leftarrow \boldsymbol{v}_a - \boldsymbol{X}_b \tilde{\boldsymbol{\beta}}_b$
15:        send $\boldsymbol{v}_b$ to Alice
16:    **end player**
17:    $i \leftarrow i + 1$
18: **end while**

---

### 2.4. Data Reconstruction

Block coordinate descent is an efficient federated learning algorithm, but can leak information about the used data set. In [4], it is explained that the attackers may reconstruct the used data set up to a rotation. From discussions with the authors of [4], we have learned that the data is better protected than by a rotation. The original data can be approximated within a quantifiable margin of error, depending on the amount of shared intermediate results. Earlier reconstruction attacks suggest that an external attacker with supplementary information might be able to mimic this approach even without access to the intermediate results. Although the design, feasibility and success of such an attack are merely hypothetical, the fact is that at this point, we cannot say to what extent the approach in [4] protects the processed data. This is one of the reasons to study a differentially private version of BCD.

This is an example of the broader problem of data privacy. It is hard, if not impossible, to measure. The reason for this is that typically no optimal attack exists. Since it is hard to know how much some optimized approach may uncover, evaluating the 'privacy' of data processing is hard. This is one of the reasons to work with theoretical upper bounds on the amount of information that is leaked. Differential privacy does precisely this. It bounds the certainty an attacker may obtain from the results of any study, regardless of the extra information or computational power the attacker may have.

**Example 1.** *Assume that an attacker has obtained a small list of possible data sets $\{(\boldsymbol{X}^{(i)}, \boldsymbol{y}^{(i)}) \mid 1 \leq i \leq n\}$, of which one is used in a deterministic study, meaning that the outcome is a function of the dataset and the label. The result of this study is a vector $\boldsymbol{\beta}$, solving (1), of weights belonging to a linear model. The attacker can simply test all possible data sets to see which ones generate the optimal weight vector $\boldsymbol{\beta}$. In this way, the attacker may determine which data set was used. This shows that deterministic methods cannot provide sufficient privacy guarantees.*

### 2.5. Differential Privacy

We begin with the standard definition of differential privacy and a localized variant [19–21]. An algorithm is $(\varepsilon, \delta)$-DP if it finds similar results for similar data sets with large probability $1 - \delta$. The similarity of the results is described by the privacy budget $\varepsilon$. In practice, this means that an attacker, who sees a certain result from the algorithm, cannot

decide which data set was used to generate the result. This implies that records in the data set remain hidden.

**Definition 1** (Differential privacy)**.** *A randomized mechanism $\mathcal{A}$ provides $(\varepsilon, \delta)$-differential privacy, if for all pairs of data sets $x_1, x_2 \in \mathcal{X}$ at distance $1 = d(x_1, x_2)$ and for any outcome y*

$$\mathbb{P}[\mathcal{A}(x_1) = y] \leq e^{\varepsilon} \, \mathbb{P}[\mathcal{A}(x_2) = y] + \delta \quad .$$

This definition provides guarantees that are unconditional on the knowledge or capabilities of the attacker. Furthermore, the parameters $\varepsilon$ and $\delta$ can be bounded from above by a variety of composition laws. The most common of these will be discussed in Section 2.6. This allows the data owner to keep track of the maximum amount of data leakage a data set has suffered.

**Example 2.** *Continuing with Example 1, assume that the list consists of two possible data sets, $\mathbf{X}_1$ and $\mathbf{X}_2$. Since there is no additional information, they are equally likely to be used. The weight vector $\boldsymbol{\beta}$ is computed using an $(\varepsilon, 0)-DP$ algorithm, where both data sets are in the universe of possible data sets. From the Definition 1 of Differential Privacy, it follows that*

$$\mathbb{P}[\mathcal{A}(\mathbf{X}_1) = \boldsymbol{\beta}] \leq e^{\varepsilon} \, \mathbb{P}[\mathcal{A}(\mathbf{X}_2) = \boldsymbol{\beta}] \quad .$$

*This implies that the likelihood that $\mathbf{X}_1$ is used is at most*

$$\frac{\mathbb{P}[\mathcal{A}(\mathbf{X}_1) = \boldsymbol{\beta}]}{\mathbb{P}[\mathcal{A}(\mathbf{X}_1) = \boldsymbol{\beta}] + \mathbb{P}[\mathcal{A}(\mathbf{X}_2) = \boldsymbol{\beta}]} \leq \frac{e^{\varepsilon}}{1 + e^{\varepsilon}} \quad .$$

*The attacker cannot learn the used data set with certainty, regardless of his computational power and additional information.*

**Definition 2** (Locally sensitive differential privacy)**.** *A randomized mechanism $\mathcal{A}$ provides $(\varepsilon, \delta)$-locally sensitive differential privacy in the data set $x_1 \in \mathcal{X}$, if for all data sets $x_2 \in \mathcal{X}$ at distance $1 = d(x_1, x_2)$ and for any outcome y*

$$\mathbb{P}[\mathcal{A}(x_1) = y] \leq e^{\varepsilon} \, \mathbb{P}[\mathcal{A}(x_2) = y] + \delta \quad .$$

Definition 1 holds for all pairs of datasets in the universe $\mathcal{X}$ at distance 1 of each other. This implies that the amount of noise added to an analysis of our dataset $\mathbf{X}$ may stem from data sets $\mathbf{D}$ and $\mathbf{D}'$ at distance 1 of each other, which are completely different from $\mathbf{X}$ and its neighbourhood. In this way, a lot of noise is added to hide the difference between $\mathbf{D}$ and $\mathbf{D}'$ while studying $\mathbf{X}$. Thus, a lot of noise has to be added to hide an absent data point, resulting in a large privacy budget with weak guarantees. Therefore, we choose to sacrifice group composition in order to obtain a closer link between the performed data analysis and the privacy budget. This results in a universe of possible data sets that is chosen with local sensitivity in mind.

Definition of a Distance

Definitions 1 and 2 make it clear that some distance on the universe of data sets must be defined. It is preferable to use concepts that make sense both in the local and the federated context. We use the following definition here. Two data sets are at a distance 1, if the sets of subjects they have data on differ by one. It thus requires suppression of an entire row of the data set. Since the data matrices should be of the same dimensions, this corresponds to having no information on someone and filling an entire row in $\mathbf{X}^{(i)}$ with zeros. This can be interpreted in the federated view too. It means that both parties remove their information about this subject from their local data. In this case, if both parties train $\varepsilon$-DP locally, this corresponds by simple composition; see Lemma 1, to $2\varepsilon$-DP in the federated setting.

### 2.6. Composition Mechanisms

The learning algorithm described in Section 3.1 consumes $\delta = 0$ and a privacy budget of $\varepsilon$ for every learning phase iteration. Using either simple composition [27,28] or advanced composition [29], it is possible to determine the consumed privacy budget for an entire protocol run.

**Lemma 1** (Simple composition). *Let $\mathcal{M}_i$ be an $(\varepsilon_i, \delta_i)$-differentially private algorithm. The sequence of algorithms*

$$\mathcal{A}(x) = (\mathcal{M}_1(x), \ldots, \mathcal{M}_T(x))$$

*is $(\sum_{i=1}^{T} \varepsilon_i, \sum_{i=1}^{T} \delta_i)$-differentially private.*

**Lemma 2** (Advanced composition). *For every $\varepsilon > 0$, $\delta \geq 0$, $\delta' > 0$ and $T \in \mathbb{N}$ the class of $(\varepsilon, \delta)$-differentially private mechanisms is $(\varepsilon', T\delta + \delta')$-differentially private under $T$-fold adaptive composition, for*

$$\varepsilon' = \varepsilon\sqrt{2T\log(1/\delta')} + T\varepsilon(e^\varepsilon - 1).$$

For sufficiently small $\delta'$ this means that the advanced composition yields better results, if

$$\sqrt{2T\log(1/\delta')} < T(2 - e^\varepsilon),$$

which leads to

$$T\log(N) < T\log(1/\delta') < \frac{T^2}{2}(2 - e^\varepsilon)^2, \tag{4}$$

since $\delta' < 1/N$. This means that advanced composition is only beneficial, if a protocol with many iterations and a small privacy budget per iteration is used and $\varepsilon < \log(2)$. For example, with $T = 5$ iterations and $\varepsilon = 0.2$, the data set may consist of at most 4 data points for advanced composition to be the better choice. Since BCD does not function with a tiny privacy budget per round, this means that we will only use simple composition.

### 2.7. Convergence

Since Algorithm 1 is iterative, an end point must be chosen. Typically, one would let the algorithm run until the result has converged, where the standard definition of convergence requires any single player to find a remainder $v_{(t)}$ in iteration $t$ that is sufficiently close to a remainder observed before,

$$\|v_{(t)} - v_{(s)}\| \leq \mathcal{B}_0 \quad \text{, for } 1 \leq s < t. \tag{5}$$

This method demands the weights to converge. However, the optimal weights may depend heavily on a single data point. It is precisely this dependence that DP tries to cap. Furthermore, when adding noise in each round, the weights will absorb some of this noise, which could lead to a series of increasing remainders, so that convergence may never occur. For these reasons (5) is not an ideal convergence definition.

At each iteration, the loss $\mathcal{L}(\beta)$ is minimized. At iteration $t$, a remainder $v_{(t)} = v_{(t-1)} - X\beta_{(t)}$ with minimum length is passed on to the next player. However, after a certain number of iterations, the benefit of an additional round will become very small. One may define that convergence is reached when the length of the remainder

$$\|v_{(t)}\|_2^2 \geq \|v_{(t-1)}\|_2^2 - \mathcal{B}_C \tag{6}$$

hardly decreases or even increases. The bound $\mathcal{B}_C$ for this would be defined at the initialization of the training. This definition is not very sophisticated, but it has the added advantage that it is directly related to the loss function, which is the objective of the training algorithm. Furthermore, it is applicable in virtually all situations. For example, it will also work in the case of increasing remainders, which may occur in a differentially private algorithm.

Rather than using convergence as stopping criterion, the experiments described here use a fixed number of $T = 5$ iterations. This makes the analysis of the algorithm and its performance simpler. Five iterations are much less than typically used in BCD. The reason for this is that extra iterations are expensive in differential privacy.

### 2.8. Code

The code used in this project is available at https://github.com/JDJ847879/dp-bcd (accessed on 12 September 2022).

## 3. Results

### 3.1. Construction of DP-BCD

If an attacker knows what function or (deterministic) computation has been performed on a data set, he may derive information about this data from the outcome. This may allow him to exclude certain data points from the data set, include other specific points or deduce relations that the data set fulfils. One of the options to limit this possibility is to hide precisely which computation has been performed. In objective perturbation [8,22–24], it is the loss function that is perturbed, preventing the attacker from knowing what computation was performed.

The algorithm presented here consists of two phases. In the first phase, all parties train a linear model on their local data set. The labels they use for this are the parts missing from the joint prediction. In the second phase, the linear models are put together to form a linear model in the federated setting. This linear model can then be published. There are two potential groups of attackers possible in this setting. During the first phase, it is the group of all other participants. At publication, it is the outside world that receives the jointly trained model. Since the group of all other participants is also part of the outside world, we will only be considering the first group when proving our privacy guarantees.

In this study, we use locally sensitive differential privacy (LSDP), as defined in Definition 2. Based on this, only data sets at distance 1 of a party's own data set are considered. Besides that, we use a small universe of possible data sets $\mathcal{X}$. It consists only of the actual data set and all data sets obtained by removing one record. We do not include possible data sets with one record more than our data set. In fact, for such a small universe, the conditions of Definitions 1 and 2 coincide.

One may argue that using the small universe based on local sensitivity to reduce the amount of noise needed while lowering the privacy budget, is in vain. This is not the case. In the transition, the privacy guarantee is shifted from absent data points with a high privacy budget to the actual data with a low privacy budget. The privacy budget is the explicit security guarantee that (LS)DP offers and as such is what users look at.

The ambition is to minimize the following 2-party loss function in both an iterative and a federated manner

$$\mathcal{L} = \left(y - \sum_{i=1}^{2} X^{(i)} \beta^{(i)}\right)^2 + \sum_{i=1}^{2} \left(\beta^{(i)}\right)^T \left(X^{(i)}\right)^T b^{(i)}. \tag{7}$$

This is the 2-party form of (2) with a perturbation term added. A ridge regression term is omitted to perform a cleaner comparison to the original BCD algorithm. However, nothing prevents such a term. In (7), each party's loss function is perturbed by the dot product of the prediction and a secret vector $b^{(i)}$, known only by party $i$.

For each party, we write that $X^{(i)} \in M_{N \times m_i}$, so there are $N$ observations of $m_i$ attributes in this party's data. It follows from our data assumption that $N > m_i$.

If the vectors $b^{(i)}$ would be sampled from a normal distribution such as (8), the perturbation term would have the added benefit that the local and federated perturbation term are of the same form. This would provide a similar perturbation term in the federated and local objective function. To avoid dimensionality problems, a different distribution is used, as explained in Remark 1.

**Remark 1.** *The vector $\boldsymbol{b} \in \mathbb{R}^N$ could be sampled from a normal distribution with density*

$$p_{naive}(\boldsymbol{b}) = \left(\frac{\varepsilon}{2\pi\xi^2}\right)^{N/2} \exp\left[-\frac{\varepsilon\|\boldsymbol{b}\|_2^2}{2\xi^2}\right] \quad . \tag{8}$$

*It is clear that the direction of the vector is uniformly sampled from the surface of the N-dimensional sphere. For its length, we want to solve for R*

$$\frac{2}{\Gamma(N/2)} \int_0^{R/(\sqrt{2}\sigma)} \mathrm{d}r\, r^{N-1} e^{-r^2} = p \qquad , with\ p \in (0,1) \quad ,$$

*which transforms into*

$$\frac{1}{\Gamma(N/2)} \int_0^{R^2/(2\sigma^2)} \mathrm{d}t\, t^{(N/2)-1} e^{-t} = p$$

*and is solved by the inverse lower incomplete gamma function. This is problematic. The high dimension pushes the vector outwards, so that the noise vectors tend to get bigger with increasing number of observations. This leads to noise vectors overwhelming the data and a remainder that is larger than the input label.*

As explained before, only the first party needs to know the labels. Afterwards, during iteration $t$, party $j$ obtains the remainder

$$\boldsymbol{v}_t^{(j)} = \boldsymbol{y} - \sum_{i=1}^{2} \boldsymbol{X}^{(i)}\boldsymbol{\beta}_t^{(i)} \quad , where \quad \boldsymbol{\beta}_t^{(i)} = \begin{cases} \sum_{s=1}^{t} \boldsymbol{\beta}_{(s)}^{(i)}, & for\ i < j \\ \sum_{s=1}^{t-1} \boldsymbol{\beta}_{(s)}^{(i)}, & for\ i \geq j \end{cases}$$

of the label that is not yet explained by party $j-1$. From now on we will suppress the sub- and superscripts when possible. The local solution is given by

$$0 = \boldsymbol{X}^T(\boldsymbol{v} - \boldsymbol{X}\boldsymbol{\beta}^* - \boldsymbol{b}) \qquad \Rightarrow \qquad \boldsymbol{\beta}^* = (\boldsymbol{X}^T\boldsymbol{X})^{-1}\boldsymbol{X}^T(\boldsymbol{v} - \boldsymbol{b}). \tag{9}$$

There are two algorithms in use in the protocol. The first is used during the learning phase to communicate the missing part of the labels. It is given by

$$\mathcal{A}_l(\boldsymbol{X}) = \boldsymbol{v} - \boldsymbol{X}\boldsymbol{\beta}^*. \tag{10}$$

The second is used in the revealing phase and is defined by

$$\mathcal{A}_r(\boldsymbol{X}) = \sum_{t=1}^{T} \boldsymbol{\beta}_{(t)}^*, \tag{11}$$

where $\boldsymbol{\beta}^*$ is in both cases defined in (9). In the special case of unperturbed learning, i.e., $\boldsymbol{b} = 0$, we call this solution $\boldsymbol{\beta}^\star$.

We start with the privacy of the learning algorithm $\mathcal{A}_l$. We sample $\boldsymbol{b} = l \cdot \boldsymbol{s}$ with $\boldsymbol{s} \in S^{N-1}$ uniformly and $l$ with density $2\sqrt{\frac{\varepsilon}{2\pi\xi^2}} \exp\left[-\frac{\varepsilon l^2}{2\xi^2}\right]$, so that

$$p_{\xi,\varepsilon}(\boldsymbol{b}) = \frac{\Gamma(\frac{N}{2})}{\pi^{\frac{N}{2}}} \sqrt{\frac{\varepsilon}{2\pi\xi^2}} \exp\left[-\frac{\varepsilon\|\boldsymbol{b}\|_2^2}{2\xi^2}\right] \tag{12}$$

Thus, the length of the perturbation vector is normally distributed and its direction is uniformly distributed. This ensures that the length of the perturbation vector is independent of the number of observations. The parameter $\xi$ is the largest allowed value of $\|\boldsymbol{v}_{out}\|$ for a successful protocol run.

The standard deviation of the length of the perturbation vector is given by $\xi/\sqrt{\varepsilon}$, where $\varepsilon$ is the privacy budget for the round and

$$\xi = \gamma \|v - X\beta^\star\|_2.$$

The parameter $\gamma > 1$ gives the maximally allowable deterioration in performance compared to the unperturbed case. It is a new parameter introduced here. It must be chosen big enough to satisfy in every iteration $t$

$$\|(v_{in,(t)} - X_i\beta^*_{(t)})_{\perp_i}\|_2^2 \leq \|v_{in,(t)} - X_i\beta^*_{(t)}\|_2^2 \leq \xi^2_{(t)}.$$

This implies that in each iteration of the protocol, the loss scaling parameter must satisfy

$$\gamma \geq \sup_{\substack{d(X,\tilde{X})=1 \\ m=\mathrm{rk}(X)=\mathrm{rk}(\tilde{X})}} \left\{ \frac{\|v - X\beta^\star\|_2}{\|v - \tilde{X}\tilde{\beta}^\star\|_2}, \frac{\|v - \tilde{X}\tilde{\beta}^\star\|_2}{\|v - X\beta^\star\|_2} \right\}. \tag{13}$$

Thus, $\gamma$ represents the cost per round of adding differential privacy to the learning algorithm. It is the multiplier of the loss with respect to the unperturbed case, where $b = 0$.

The probability that two databases $X_1, X_2$ of full rank at a distance 1 of each other yield the same output vector $v_{out} = v_{in} - X_1\beta^*_1 = v_{in} - X_2\beta^*_2$ is, according to (9), given by

$$\begin{aligned}
\frac{\mathbb{P}[\mathcal{A}_l(X_1) = v_{out}]}{\mathbb{P}[\mathcal{A}_l(X_2) = v_{out}]} &= \frac{\mathbb{P}[0 = X_1^T(v_{out} - b_1)]}{\mathbb{P}[0 = X_2^T(v_{out} - b_2)]} = \frac{\mathbb{P}[b_1 \in \ker(X_1^T) + v_{out}]}{\mathbb{P}[b_2 \in \ker(X_2^T) + v_{out}]} \\
&= \frac{\mathbb{P}[b_1 \in \ker(X_1^T) + v_{1,\perp_1}]}{\mathbb{P}[b_2 \in \ker(X_2^T) + v_{2,\perp_2}]} \\
&= \frac{\exp[-\frac{\varepsilon}{2\xi^2}\|v_{1,\perp_1}\|_2^2]}{\exp[-\frac{\varepsilon}{2\xi^2}\|v_{2,\perp_2}\|_2^2]} \leq e^\varepsilon
\end{aligned} \tag{14}$$

Here, we have decomposed $v_{out} = v_{1,\ker_1} + v_{1,\perp_1} = v_{2,\ker_2} + v_{2,\perp_2}$ into parts inside the kernel and perpendicular to it. Note that the decomposition for $X_1$ is different from that for $X_2$. For the probabilities, it suffices that

$$\exp[-\alpha(v_\perp + \sum_j \lambda_j w_j)^2] = \exp[-\alpha v_\perp^2] \cdot \prod_j \exp[-\alpha \lambda_j^2],$$

where $\{w_j\}$ is an orthonormal basis for the kernel. Note that the parts inside the kernel can only stem from $v_{in}$. Since both matrices are of full rank, their kernels have the same dimensions and selecting a vector out of them is equally likely. For the perpendicular parts, a standard argument can be used. Using (13), the final inequality follows from

$$\|v_{i,\perp}\|_2^2 \leq \|v_{out}\|_2^2 = \|v_{in} - X_i\beta^*_i\|_2^2 \leq \xi^2.$$

For the revealing phase, a very similar argument works. Instead of the missing labels, it is now the weights that are communicated. The privacy loss for revealing a single $\beta^*_{(t)}$ is computed by

$$\frac{\mathbb{P}[\mathcal{A}_r(X_1) = \beta^*_{(t)}]}{\mathbb{P}[\mathcal{A}_r(X_2) = \beta^*_{(t)}]} = \frac{\mathbb{P}[0 = X_1^T(v_{in,(t)} - X_1\beta^*_{(t)} - b_{1,(t)})]}{\mathbb{P}[0 = X_2^T(v_{in,(t)} - X_2\beta^*_{(t)} - b_{2,(t)})]}$$

$$= \frac{\mathbb{P}[b_{1,(t)} \in \ker(X_1^T) + (v_{in,(t)} - X_1\beta^*_{(t)})_{\perp_1}]}{\mathbb{P}[b_{2,(t)} \in \ker(X_2^T) + (v_{in,(t)} - X_2\beta^*_{(t)})_{\perp_2}]}$$

$$\leq \frac{\exp[-\frac{\varepsilon}{2\xi^2_{(t)}} \|(v_{in,(t)} - X_1\beta^*_{(t)})_{\perp_1}\|_2^2]}{\exp[-\frac{\varepsilon}{2\xi^2_t} \|(v_{in,(t)} - X_2\beta^*_{(t)})_{\perp_2}\|_2^2]} \leq e^\varepsilon. \tag{15}$$

From simple composition, Lemma 1, it follows that revealing the weights $\sum_{t=1}^T \beta^*_{(t)}$ consumes at most a privacy budget of $T\varepsilon$.

To demand that observations should generate a full rank matrix is a minor demand. If it were not the case, a certain attribute could be predicted perfectly by the other attributes. Hence, it could be removed from the database to generate a full rank matrix again. Furthermore, it is not necessary for the proof to work with full rank matrices. They should only be of equal rank.

The complete 2-party algorithm DP-BCD is shown in Algorithm 2. A generalization to more parties is straightforward.

---

**Algorithm 2** Differentially private 2-party block coordinate descent algorithm

---

1: $\varepsilon' > 0$, $T \in \mathbb{N}$ and $\gamma > 1$
2: Alice and Bob initiate $\beta_a \leftarrow \mathbf{0}$ and $\beta_b \leftarrow \mathbf{0}$, respectively.
3: Alice initiates $v_b \leftarrow y$
4: **for** $t \in \{1, \dots, T\}$ **do**
5:     **player** Alice **do**
6:         $\xi_a = \gamma \|v_b - X_a\beta_a^\star\|_2$
7:         $b_a \sim p_{\xi_a, \varepsilon'/(2T)}$
8:         $\tilde{\beta}_a \leftarrow (X_a^T X_a)^{-1} X_a^T(v_b - b_a)$
9:         $\beta_a \leftarrow \beta_a + \tilde{\beta}_a$
10:        $v_a \leftarrow v_b - X_a\tilde{\beta}_a$
11:        **if** $\|v_a\|_2 \leq \xi_a$ **then**
12:           send $v_a$ to Bob
13:        **else**
14:           abort
15:        **end if**
16:     **end player**
17:     **player** Bob **do**
18:         $\xi_b = \gamma \|v_a - X_b\beta_b^\star\|_2$
19:         $b_b \sim p_{\xi_b, \varepsilon'/(2T)}$
20:         $\tilde{\beta}_b \leftarrow (X_b^T X_b)^{-1} X_b^T(v_a - b_b)$
21:         $\beta_b \leftarrow \beta_b + \tilde{\beta}_b$
22:        $v_b \leftarrow v_a - X_b\tilde{\beta}_b$
23:        **if** $\|v_b\|_2 \leq \xi_b$ **then**
24:           send $v_b$ to Alice
25:        **else**
26:           abort
27:        **end if**
28:     **end player**
29: **end for**
30: Alice sends $\beta_a$ to Bob.
31: Bob sends $\beta_b$ to Alice.
32: Alice and Bob publish $(\beta_a, \beta_b)$.

---

Utility Bound

During the protocol run, the participants must check in every iteration whether the loss increase is less that a factor $\gamma$, as demanded in (13). If this is not the case, the protocol will be aborted by the participants, because a model with sufficient utility cannot be trained. Hence, at every single iteration the sum of squared errors, which is the unperturbed loss, is bounded by

$$\|v - X\beta^*\|_2^2 \leq \gamma^2 \|v - X\beta^\star\|_2^2.$$

This information can be used in another way. It is directly related to the utility loss and provides an upper bound for the utility loss. In a protocol run with $k = 2$ parties and $T$ iterations, the sum of squared errors is at most a factor $\gamma^{2kT}$ larger than in the unperturbed case. If we denote with $f_*$ the differentially private predictions and with $f_\star$ those without DP, then we observe that the utility measure

$$R^2 = 1 - \frac{\|y - f_*\|_2^2}{Var_y} \geq 1 - \gamma^{2kt} \frac{\|y - f_\star\|_2^2}{Var_y} = 1 - \gamma^{2kt}(1 - R_\star^2). \tag{16}$$

This shows that we obtain a utility guarantee along with the privacy guarantee. The additional utility loss is bounded by parameters that can be set before the start of the protocol.

This proves the following theorem.

**Theorem 1.** *The linear regression of **y**, held by Alice, against the data $(X_a, X_b)$ can be approximated by Algorithm 2, provided that $\mathrm{rk}(X_a) = m_a$ and $\mathrm{rk}(X_b) = m_b$ are of full column rank and contain N data points, where $N > m_a$ and $N > m_b$. For $T \in \mathbb{N}$, $\varepsilon' > 0$ and $\gamma > 1$ it is an $\varepsilon'$-differentially private algorithm. Furthermore, the utility is bounded from below by*

$$R^2 \geq 1 - \gamma^{4T}(1 - R_\star^2),$$

*where $R_\star^2$ is the utility of the block coordinate algorithm without differential privacy (Algorithm 1).*

*3.2. Experiments on Synthetic Data*

In order to quantify the performance of DP-BCD simulations with synthetic data are performed. We use standard normally distributed data and normally distributed $\beta$ parameters ($\mu = 2$, $\sigma = 1.5$). In the baseline scenario, there are nine predictors, with a correlation of 0.3, $N = 1000$, $R^2 = 0.3$, $\varepsilon = 1$, and $\gamma = 1.2$ with two parties. Because preliminary analyses have indicated that five iterations is a favourable cut-off in the trade-off between privacy and noise-accumulation, this is the number of iterations used.

For comparison with this baseline scenario, each of the following factors are varied separately: the sample size $N \in \{100, 250, 1000, 5000, 10,000\}$, the correlation between predictors $\{0.1, 0.3, 0.5\}$, $R^2 \in \{0.1, 0.3, 0.8\}$, $\varepsilon \in \{0.1, 0.3, 0.5, 0.8, 1.0, 1.5, 2.5, 10\}$, and $\gamma \in \{1.15, 1.25, 1.5, 1.8, 2, 2.5, 3\}$. The $\gamma$ values are chosen big enough to avoid an abortion of the protocol run with high probability. For low values of $\gamma$, the algorithm may terminate (see Algorithm 1, because $\gamma$ is too low. This could lead to an unbalanced comparison between scenarios where the $\gamma$ is sufficiently high and those where the algorithm could not carry out all iterations for each repetition. Each of the variations is repeated 500 times with the exception of the sample size experiment, which is repeated 100 times per variation. At every iteration, a different data set ($X$ and $y$) is generated. In experiments where the privacy parameters $\varepsilon$ and $\gamma$ are varied, different $\beta$ parameters are generated for every iteration.

To evaluate the utility, two results are considered. These are the $R^2$ and the $\beta$ estimates. These outcomes are also generated in the centralized setting and using BCD without differential privacy. Because the results for these two algorithms are practically identical, we only compare it to the centralized results. For several scenarios, we compute the average absolute proportional distance (AAPD) for these $\beta$ estimates. For $r$ repetitions of a scenario with $m$ predictors, the corresponding AAPD is defined as

$$\text{AAPD} = \frac{\sum_{i=1}^{r} \|\boldsymbol{\beta}^* - \boldsymbol{\beta}^\star\|_1}{rm}.$$

### 3.2.1. Impact of Privacy Parameters

The impact of $\gamma$ and $\varepsilon$ on the $\beta$ and $R^2$ estimates is non-linear. We find that $\gamma$ has a stronger impact on $R^2$ than $\varepsilon$. From Figure 1b, it can be observed that the bound for $R^2$ decreases significantly with $\gamma$. However, for the synthetic data, the expected decrease is not nearly as steep as its bound. For example, for $\gamma = 3$, the average $R^2$ is approximately $-2.5$, whereas it is bounded by $-2.44 \times 10^9$. Although the results are in line with Theorem 1, the bound can be almost meaningless for large values of $\gamma$.

The $\beta$ estimates grow closer to the BCD results as $\varepsilon$ increases, which is in line with the expectation. Table 1 shows that for $\varepsilon = 1$, the $\beta$ estimates deviate 47% from the centralized $\beta$ parameters on average. For $\gamma = 1.15$ (the lowest tested value), the $\beta$ estimates deviate 295% from the centralized setting, but note that this is for $\varepsilon = 1$, see Table 2. For higher values of $\varepsilon$, the estimates are closer to the centralized $\beta$ parameters, though still differing by up to 47%. As a reference, the average and median deviation after five iterations for BCD without DP are practically zero.

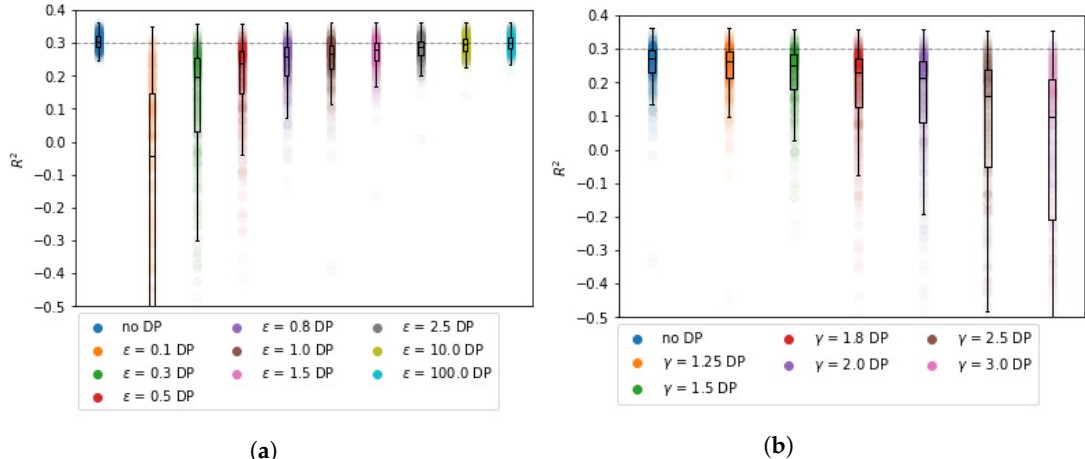

(**a**)            (**b**)

**Figure 1.** $R^2$ of DP-BCD in artificial test data. (**a**) $R^2$ after DP-BCD as a function of $\varepsilon$; (**b**) $R^2$ after DP-BCD as a function of $\gamma$.

**Table 1.** Mean and median proportional absolute error of $\beta$ estimates compared to centralized setting, over $\varepsilon$ after five iterations for $\gamma = 1.2$.

| $\varepsilon$ | Mean | Median |
|:---:|:---:|:---:|
| 0.2 | 1.50 | 10.09 |
| 1.0 | 0.67 | 4.37 |
| 2.0 | 0.47 | 3.08 |
| 3.0 | 0.38 | 2.51 |
| 5.0 | 0.30 | 1.94 |
| 10.0 | 0.21 | 1.37 |
| 20.0 | 0.15 | 0.97 |

**Table 2.** Mean and median proportional absolute error of $\beta$ estimates compared to centralized setting, over $\gamma$ after five iterations for $\varepsilon = 1$.

| $\gamma$ | Mean | Median |
|---|---|---|
| 1.15 | 2.95 | 0.45 |
| 1.25 | 3.21 | 0.49 |
| 1.50 | 3.86 | 0.59 |
| 1.80 | 4.64 | 0.71 |
| 2.00 | 5.17 | 0.79 |
| 2.50 | 6.51 | 0.98 |
| 3.00 | 7.87 | 1.18 |

### 3.2.2. Impact of $R^2$

As $R^2$ increases in the data-generating model, more predictive power is preserved with DP-BCD as well, see Figure 2. The precision and bias with which the $\beta$ parameters can be estimated are also significantly impacted by $R^2$ in the data generating model, see Table 3.

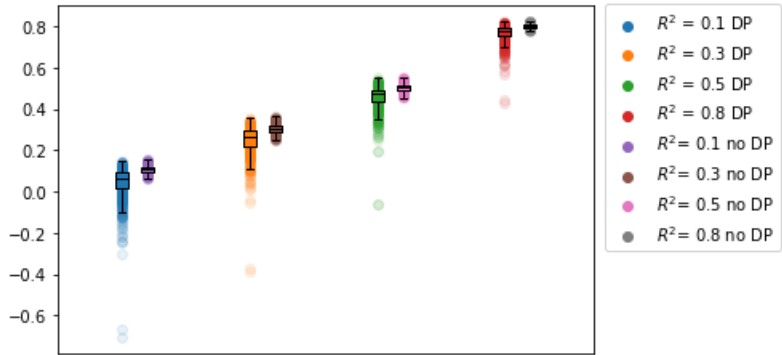

**Figure 2.** $R^2$ after DP-BCD as a function of $R^2$ in the data-generating model.

**Table 3.** Mean and median proportional absolute error over $R^2$.

| $R^2$ | Mean | Median |
|---|---|---|
| 0.1 | 4.72 | 0.85 |
| 0.3 | 3.08 | 0.47 |
| 0.5 | 1.88 | 0.33 |
| 0.8 | 1.02 | 0.21 |

### 3.2.3. Impact of Correlation

The impact of the correlation on the utility of the learned model can be seen in Figure 3. As expected, the average $\beta$ error increases with the correlation between predictors. This can be observed in the wider sampling distribution in Table 4. This is to be expected for an implementation of DP, since more noise must be added to hide the outliers in the data. For very high correlations, the average $\beta$ parameters differ as well, which means that the estimates are biased. The $R^2$, however, remains unaffected by this parameter, though it is lower than with the BCD algorithm.

As studied by [4], strongly correlated data require more iterations for accurate parameter estimation. In fact, for highly correlated data with over 25 variables, hundreds of iterations can be required for convergence of the weights. In a differential privacy setting, this may consume vast privacy budgets or yield poor results due to noise accumulation.

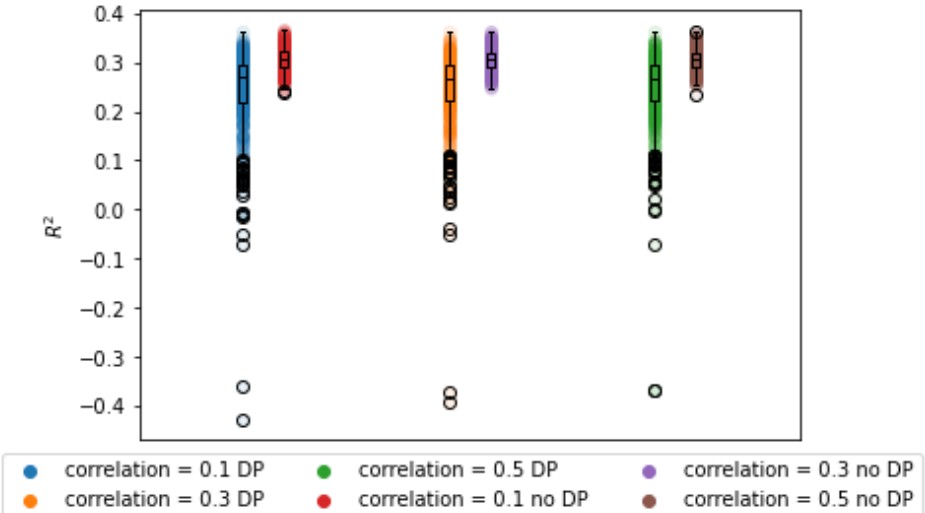

**Figure 3.** $R^2$ after DP-BCD as a function of the correlation between predictors in the data-generating model.

**Table 4.** Mean and median proportional absolute error over the correlation between predictors.

| Correlation | Mean | Median |
|:-----------:|:----:|:------:|
| 0.1 | 0.56 | 0.27 |
| 0.3 | 1.06 | 0.45 |
| 0.5 | 3.11 | 0.69 |

### 3.2.4. Impact of Sample Size

Sample size is well known to have a large impact on the performance of differentially private model, see Figure 4. As can be observed from Table 5, the $\beta$ error steadily decreases with the sample size. Furthermore, the $R^2$ distribution grows closer to the centralized results.

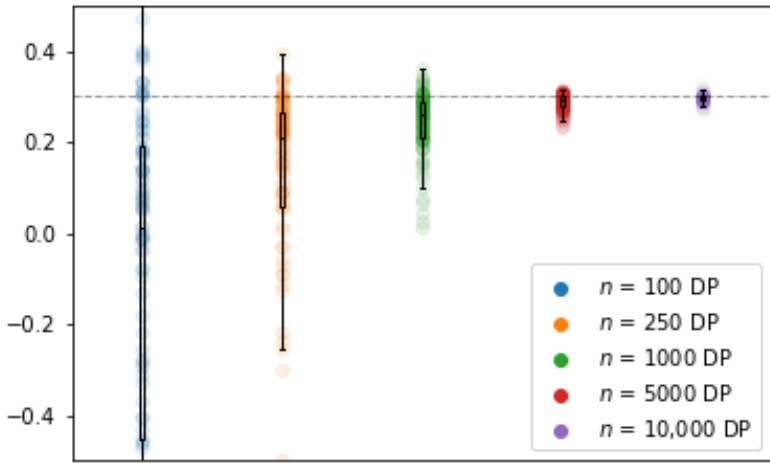

**Figure 4.** $R^2$ after DP-BCD as a function of $N$.

**Table 5.** Mean and median proportional absolute error over *N*.

| N | Mean | Median |
|---|---|---|
| 100 | 7.68 | 1.46 |
| 250 | 3.82 | 0.95 |
| 1000 | 2.30 | 0.47 |
| 5000 | 0.85 | 0.22 |
| 10,000 | 0.59 | 0.15 |

### 3.3. Evaluation with Real-World Data

We run experiments with two real-world data sets: a forest fires data set by [25], which was used by [4] and a Garment Industry employee productivity data set by [26]. For both data sets, we computed the average coefficients, using $\gamma = 1.2$, $\varepsilon = 1, 10$, $T = 5$ iterations and repeated the experiment 1000 times. We plot the the 2.5th and 97.5th percentiles and compare this to the parameter estimates for the centralized analysis. In addition, $R^2$ is computed in every iteration and plotted for the $\varepsilon$ values of 0.2, 1, 2, 5, and 10. For both data sets we use two parties.

#### 3.3.1. Evaluation with Forest Fires Data

The forest fires data set contains 517 records with 12 predictors containing meteorological and other information to predict the burned area of forest fires. A total of 27 predictors were used in the regression analysis, with the variables pertaining to the month and day transformed to dummy variables.

The plot in Figure 5 shows a plot similar to Figure 5 of [4] using the same data and parties. We also plot the parameter estimates for the centralized analysis (which [4] was demonstrated to be almost identical to BCD with 450 iterations).

For a relatively small privacy budget of $\varepsilon = 1$, the average coefficients are similar to those from the centralized setting. For $\varepsilon = 10$, the distributions are narrower, which is in line with the synthetic data results. The closeness of the sampling distributions to the centralized setting is likely affected by the low correlations between the predictors (with an absolute average and median of 0.08 and 0.05, respectively).

The $R^2$ values are quite low, due to the fact that the centralized $R^2$ is only 0.07. Because $R^2$ values for DP-BCD are generally lower than BCD, all median $R^2$ values are negative for the forest fires data. The y-axis in Figure 6 is cut off at $-0.5$, because negative $R^2$ values are not informative, but that for $\varepsilon = 1.0$ and $\varepsilon = 2.0$ the median $R^2$ values are $-4.07$ and $-0.94$, respectively. Thus, for a centralized model that already has low predictive power, adding differential privacy generally results in a complete loss of predictive power.

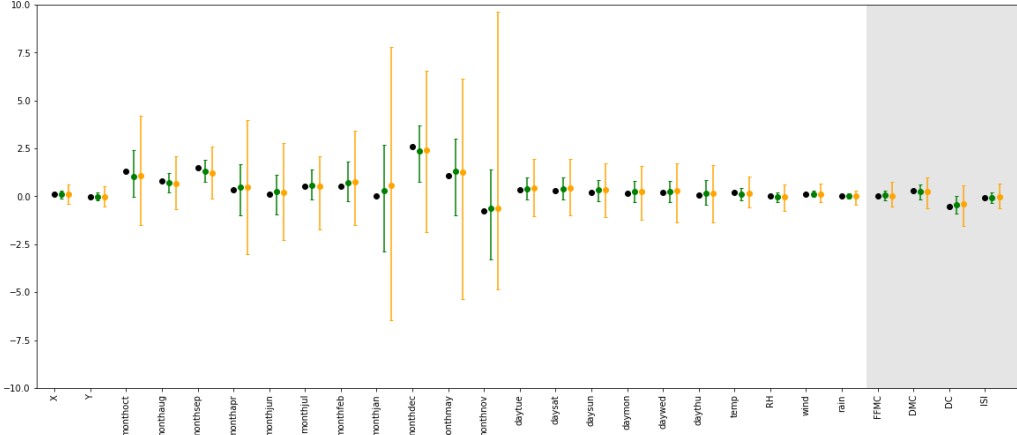

**Figure 5.** Centralized parameter estimates (black) for forest fire analysis, with average coefficients and 95% confidence intervals for $\varepsilon = 1$ (orange) and $\varepsilon = 10$ (green), for $\gamma = 1.2$, 1000 repetitions. Parties are separated with background shading.

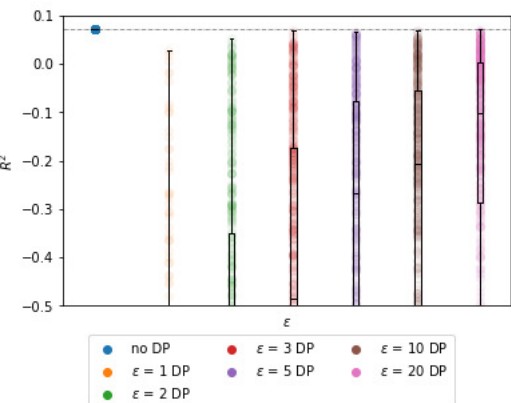

**Figure 6.** $R^2$ over $\varepsilon$ for the forest fires data set (5 iterations, 100 repetitions, and $\gamma = 1.2$).

### 3.3.2. Evaluation with Garment Employee Productivity Data

We have also tested the algorithm with data from the Garment Industry by [26]. This data set contains 1197 employee records, with 15 predictors for employee productivity (on a continuous 0–1 scale). We removed the variables wip (to avoid missing values) and date (for a simpler regression problem) and used dummy variables for department and date. This data set also has quite low correlations, with a median correlation of 0.03. Figure 7 depicts the distribution of variables between the parties.

On average, the $\beta$ estimates are close to the centralized analysis, although they do differ with a single run (see Figure 7). The effect of $\varepsilon$ is similar to that for the forest fires analysis. The distribution is narrower for $\varepsilon = 10$.

With respect to $R^2$, Figure 8 depicts that the relation between $\varepsilon$ and $R^2$ is similar to those observed for the synthetic data and forest fires data. However, compared to the forest fires data set, more predictive power is preserved, which is related to centralized $R^2$ of 0.24, This can be observed from the median $R^2$ values, which are both higher and closer to the centralized results.

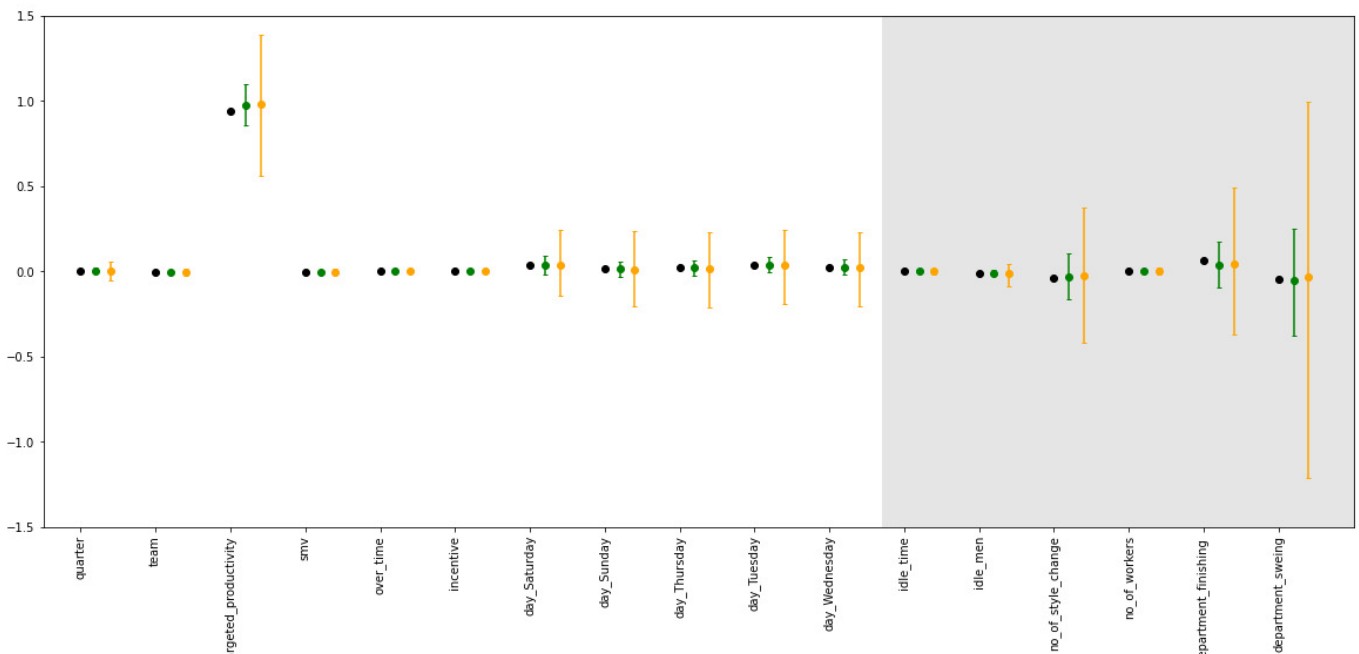

**Figure 7.** Centralized parameter estimates (black) for regression analysis with Garment employee productivity data set, with average coefficients and 95% confidence intervals for $\varepsilon = 1$ (orange) and $\varepsilon = 10$ (green), for $\gamma = 1.2$, 1000 repetitions. Parties are separated with background shading.

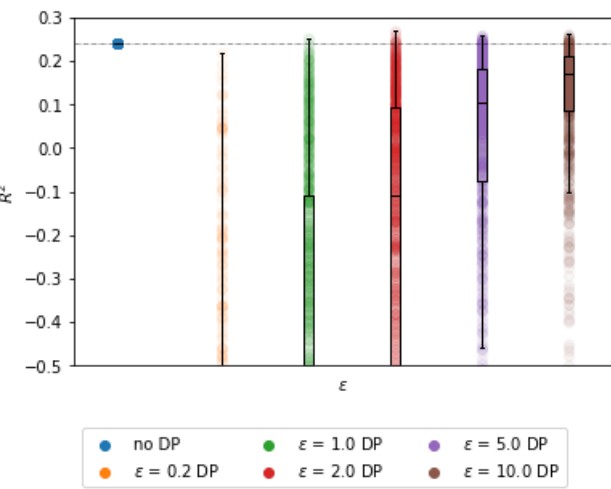

**Figure 8.** $R^2$ over $\varepsilon$ for the Garment employee productivity data set (5 iterations, 1000 repetitions, and $\gamma = 1.2$).

## 4. Discussion

In this paper, we have constructed and tested the DP-BCD algorithm. We have demonstrated that it comes with a utility bound, that bounds the loss of $R^2$ as a function of the privacy parameters and the $R^2$ of the BCD solution. This is a new concept in DP learning. However, in its current form its practical relevance is very small, since the bound is too wide. The lower bound in this form deteriorates quickly with an increasing number of parties and iterations.

The simulations in Sections 3.2 and 3.3 demonstrate that the weights obtained with DP-BCD are similar to BCD, also for correlated data. Nonetheless, the predictive power is lower, especially for problems with low $R^2$. Nonetheless, the median $R^2$ values observed are similar to the BCD, albeit with larger deviations and some outliers. We find that the predictive power is considerably lower for small values of $\varepsilon$, high values of $\gamma$ or small sample sizes, provided that $\gamma$ is chosen big enough not to abort.

Both $\gamma$ and $\varepsilon$ have a strong impact on the predictive power. Therefore, the $\gamma$ value should be set as low as possible, as there is no benefit to having a high $\gamma$. With respect to $\varepsilon$, the algorithm retained predictive power even for single-digit privacy budgets. Though not incorporated in the simulation, the number of parties is also expected to impact $R^2$, as it makes the BCD procedure more challenging and has a significant impact on the utility bound.

Unbiased estimation of $\beta$ parameters is a more challenging task than retaining predictive power. With the current procedure, this is not feasible with the amount of noise required. Particularly for highly correlated variables, the number of iterations may exceed the point where the increased precision as a result of iterations is overshadowed by the accumulated noise. For large sample sizes and large values of $\varepsilon$, it is possible to obtain $\beta$ parameters similar to the BCD procedure. This was also visualized in the forest fire analysis, where $\varepsilon = 20$ led to parameter estimates closer, though not identical, to the centralized and BCD setting.

A fixed number of iterations has been used in the experiments. In this way, a clearer presentation of the performance of DP-BCD can be given. However, a convergence criterion, as described in Section 2.7, makes it possible to explicitly decide each round whether the improved utility is worth the consumed privacy budget. In this way, algorithms with better performance in terms of privacy budget and utility can be constructed.

The problem of federated linear regression on vertically partitioned data is also studied in [15]. Based on the used techniques, it is our estimate that their solution can provide a higher utility on average, since it only requires a single round of noise addition. On the other hand, the use of secure multiplication techniques will probably lead to a longer

learning phase, we expect our solution to be faster. Since we had no access to their code, we can only compare the solutions qualitatively.

The approach chosen in this article may work as well for logistic regression. The BCD algorithm can also be used for logistic regression [4] on vertically partitioned data. It has been demonstrated that objective perturbation works well for logistic regression [23]. However, it is not clear whether it is possible to provide a utility guarantee of a similar nature for logistic regression.

## 5. Conclusions

In this article, we have presented a differentially private extension of the block coordinate descent algorithm for a single label owner, called DP-BCD. We demonstrate that in scenarios where privacy concerns or regulations limit collaborative opportunities, DP-BCD can be used to enable multi-party collaboration, with strict privacy bounds. The algorithm can be used for linear regression analysis of vertically partitioned data. Our construction applies objective perturbation in combination with a small universe of possible data sets following from local sensitivity. In this way, we are able to generate models with both comparable predictive power as BCD and single digit privacy budgets. Furthermore, the set-up allows for a theoretical utility bound that gives a lower bound for the $R^2$ of the differentially private version in terms of that of the original algorithm.

The acceptable performance loss of DP-BCD compared to BCD is parametrized by a new parameter $\gamma$. It allows parties to agree on both a privacy and a utility goal. A direct consequence of this is that DP-BCD comes with theoretical utility guarantees.

Experiments indicate that DP-BCD performs particularly well in settings where the data has a high $R^2$, meaning that the data contains a lot of explanatory power. Furthermore, the low number of iterations used benefits data sets with little correlation. For the real-world data sets, we find that the obtained weights are similar on average, although the $R^2$ is lower.

**Author Contributions:** J.d.J.: conceptualization, methodology, formal analysis, writing—original draft preparation. B.K.: conceptualization, software, validation, writing—review and editing. S.K.: investigation, writing—original draft preparation, visualization. All authors have read and agreed to the published version of the manuscript.

**Funding:** This research received no external funding.

**Institutional Review Board Statement:** Not applicable.

**Informed Consent Statement:** Not applicable.

**Data Availability Statement:** Publicly available datasets were analyzed in this study. This data can be found here: [https://archive.ics.uci.edu/ml/datasets/Forest+Fires] and [https://archive.ics.uci.edu/ml/datasets/Productivity+Prediction+of+Garment+Employees].

**Acknowledgments:** The authors would like to thank Ásta Magnúsdóttir and Savvina Daniil for their contributions to the project.

**Conflicts of Interest:** The authors declare no conflict of interest.

## Abbreviations

The following abbreviations are used in this manuscript:

| | |
|---|---|
| AAPD | Average absolute proportional distance |
| BCD | Block coordinate descent |
| DP | Differential privacy |
| DP-BCD | Differentially private block coordinate descent |
| LSDP | Locally sensitive differential privacy |

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
