# Peer review of "Differentially Private Block Coordinate Descent for Linear Regression on Vertically Partitioned Data"

_jcp, doi:10.3390/jcp2040044_

Round 1

Reviewer 1 Report

This paper utilizes differential privacy (DP) in the block coordinate descent (BCD) algorithm to achieve privacy-preserving federated learning. The author also carries out many experiments to verify the practicality of DP. The problem is described clearly and the method is easy to follow. However, I have the following concerns.

1. The references are insufficient. There are a lot of DP-based federated learning algorithms. More related work should be briefly introduced and compared with the proposed method. E.g., the work "Low-Latency Federated Learning Over Wireless Channels With Differential Privacy" should be compared.

2. In the experiment section, the author demonstrate the impact of different level of DP on the trained model. However, the privacy issue is not evaluated. Since the aim of applying DP is to fulfill privacy-preserving, an evaluation about data leakage on different DP level should also be presented.

Author Response

Dear reviewer,

Thank you for your response and suggestions to improve our submission.  We
have carefully reviewed the comments and have revised the manuscript accordingly. Our responses are given in a point-by-point manner below.

We have provided an additional experiment to support our conclusions. And we have rewritten parts of our article to make the presentation clearer.

1. The references are insufficient. There are a lot of DP-based federated learning algorithms. More related work should be briefly introduced and compared with the proposed method. E.g., the work "Low-Latency Federated Learning Over Wireless Channels With Differential Privacy" should be compared.

Thank you for pointing this out to us. This was indeed the case and we have provided 16 additional articles describing related work and relevant results we use.

2. In the experiment section, the author demonstrate the impact of different level of DP on the trained model. However, the privacy issue is not evaluated. Since the aim of applying DP is to fulfill privacy-preserving, an evaluation about data leakage on different DP level should also be presented.

It is hard and virtually impossible to test how much information can be retrieved from a learning algorithm. This is the reason to use theoretical privacy guarantees of a mathematical nature, where the privacy gurantee is quantified by the DP parameters. However, this became insufficiently clear in the original submission. To this end we have added explanations and examples to sections 2.4 and 2.5.

We hope the revised manuscript will better suit the Journal of Cybersecurity and Privacy, but are happy to consider further revisions. We thank you for your interest in our research.

Yours faithfully,
Jins de Jong

Reviewer 2 Report

This paper needs serious revisions. Following are my major concerns:

·         The abstract and conclusion are not written properly; the authors need to rewrite it.

·         The motivation and contribution of this work needs to be highlighted in proper manner in the introduction section to make it clearer for the readers to comprehend the main theme targeted in this article.

·       The authors should highlight their contributions in a proper manner by emphasizing on how their work is different from other articles? 

·         The authors should add a Related work section to summarize the existing literature and show how their work is different from the existing literature.

·       Section 2 and 3 are the core parts of this paper but it is not presented properly since the current organization/presentation is confusing. The authors are advised to present these in proper manner.

·         The algorithms need more discussion and also cost analysis should be presented for the algorithms.

·         The federated architecture needs more discussion and explain how the results are aggregated?

·         The proposed model needs to be tested with more data from different domain and also the results needs a comparison with recently published studies from top journals and conferences.

·         More results and discussion are required to validate the performance of the proposed model.

·         The paper needs rewriting and language polishing since it has many typos and English mistakes.

Author Response

Dear reviewer,

Thank you for your response and suggestions to improve our submission.  We have carefully reviewed the comments and have revised the manuscript accordingly. Our responses are given in a point-by-point manner below.

- The abstract and conclusion are not written properly; the authors need to rewrite it.
This we have done.

- The motivation and contribution of this work needs to be highlighted in proper manner in the introduction section to make it clearer for the readers to comprehend the main theme targeted in this article.
It would be a pity, if interested readers were lost during the introduction. Therefore, we have overhauled the introduction and put more emphasis on the context of our work.

- The authors should highlight their contributions in a proper manner by emphasizing on how their work is different from other articles? 
We have rewritten the "Our contributions" section to emphasize the new ideas we present.

- The authors should add a Related work section to summarize the existing literature and show how their work is different from the existing literature.
Thank you for pointing this out to us. This was indeed the case and we have provided 16 additional articles describing related work and relevant results we use.

- Section 2 and 3 are the core parts of this paper but it is not presented properly since the current organization/presentation is confusing. The authors are advised to present these in proper manner.
We have restructered sections 2 and 3 and have provided more explanation of the concepts. 

- The algorithms need more discussion and also cost analysis should be presented for the algorithms.
We have added a description of the BCD algorithm in section 2,3 to provide the reader with more intuition on the algorithm. We have also tried to present the various ingredients of the DP-BCD algorithm in a clearer way.
A cost analysis of the algorithm has not been provided, since it is composed of lightweight techniques and only uses a very small number of iterations. A comparison of runtime with the algorithm from "Achieving Differential Privacy in Vertically Partitioned Multiparty Learning" would indeed be interesting, but is out of reach, since we have no access to their source code.

- The federated architecture needs more discussion and explain how the results are aggregated?
In section 2.1 of the revised manuscript an explanation of the federated context is provided. The aggregation of the results was implicit in the original manuscript. It has been added explicitly in section 2.1 and algorithm 2.

- The proposed model needs to be tested with more data from different domain and also the results needs a comparison with recently published studies from top journals and conferences.
We have added another experiment from the clothing industry to support our conclusions. The problem considered in "Achieving Differential Privacy in Vertically Partitioned Multiparty Learning" comes closest to ours. In the discussion we explain now why a direct comparison is not feasible. Their source code is not available, so that a comparison on both utility and runtime is impossible.

- More results and discussion are required to validate the performance of the proposed model.
We thank the reviewer for notification. We believed a shortcoming of our paper was the lack of a real-world experiment with a higher R^2 in the central setting. The added experiment on the garment industry data set has filled this gap and supports our conclusions. We have tried to perform the experiments that are key to understand the behaviour of DP-BCD. Undoubtedly, there are many more interesting experiments that may point out interesting characteristics of this learning method. We hope that many researcher will use the provided source code to this end and publish additional insights about DP-BCD.

- The paper needs rewriting and language polishing since it has many typos and English mistakes.
Thank you for pointing this out to us.

We hope the revised manuscript will better suit the Journal of Cybersecurity and Privacy, but are happy to consider further revisions. We thank you for your interest in our research.

Yours faithfully,
Jins de Jong

Round 2

Reviewer 2 Report

I have one more question: From the onset it seems that the proposed architecture is fixed one; what of more servers and more data are added?

Author Response

Dear reviewer,

Thank you for your quick response.

- I have one more question: From the onset it seems that the proposed architecture is fixed one; what of more servers and more data are added?
This is indeed the case, but has not been made explicit. That is why we have added an explanantion of these assumption to section 2.1. We have also added a few notes on the situation of changing data or participants.

We are grateful for and happy to answer your questions. Thank you for your contributions to our work.

Yours faithfully,
Jins de Jong